# The origin and evolution of IncF33 plasmids based on large-scale data sets

Guolong Gao,[1,2] Wanyun He,[1,2] Yanxiang Jiao,[1,2] Zhongpeng Cai,[1,2] Luchao Lv,[1,2] Jian-Hua Liu[1,2]

**ABSTRACT** To comprehensively understand the formation and evolution of IncF33 plasmids, a global collection of whole-genome sequences of 863 strains positive for IncF33 plasmid replicons was analyzed. The results showed that the IncF33 plasmids were mainly identified in *Enterobacterales*, of which *Escherichia coli* (86.44%) was the dominant host, followed by *Salmonella* (8.57%) and *Klebsiella pneumoniae* (3.48%). *Salmonella* ST11 and *K. pneumoniae* ST11 were common and mostly from humans. IncF33 plasmids were found worldwide but prevalent in Chinese farm animals, predominantly carrying antibiotic resistance genes such as $bla_{CTX-M-55}$, $fosA3$, $bla_{TEM-1B}$, $rmtB$, and $floR$. Comparative genomics analysis of 103 complete IncF33 plasmid sequences showed highly similar backbones except for 16 lacking partial backbone fragments. Variable regions were diverse, containing various antibiotic resistance genes, insertion sequences, or other plasmid fragments. They can be roughly divided into two sublines based on the production of different CTX-M enzymes. Similar IncF33 plasmids from different countries were identified. Some early IncF33 plasmids lacked part of their leader regions, which showed over 99% homology with early F2:A-:B- plasmids, indicating that leader regions of IncF33 likely came from F2:A-:B- plasmids. In addition, IncF33 plasmids cointegrating with other types of plasmids to form new cointegrate plasmids are increasing, making them more efficient in their dissemination and persistence in *Enterobacterales*, which could pose a significant threat to global public health.

**IMPORTANCE** Plasmids that capture multiple antibiotic resistance genes are spreading widely, leading to the emergence and prevalence of multidrug-resistant bacteria. IncF33 plasmids are a newly emerged plasmid type highly prevalent in animal-source *Enterobacterales* in China, and they are important vectors for transmitting several clinically important antibiotic resistance genes. The study revealed that the IncF33 plasmid is mainly prevalent in China animal-derived *Escherichia coli* and has the potential for cointegration and intercontinental dissemination. Therefore, it is crucial to enhance surveillance and control measures to limit the spread of IncF33 plasmids and their associated antibiotic resistance genes.

**KEYWORDS** evolution, plasmid, comparative genomics

The emergence and spread of multidrug-resistant bacteria pose a serious threat to global public health and have attracted increasing attention on a worldwide scale (1). Particularly, multidrug-resistant *Enterobacterales* are of great concern. The horizontal spread of antibiotic resistance genes (ARGs) is the main reason for the acquisition of resistance in *Enterobacterales*. Of these, plasmids are crucial vehicles for the effective transmission of ARGs.

Currently, there are 28 known plasmid types in *Enterobacterales* distinguished by PCR-based replicon typing (2). Frequently reported plasmid families include IncF, IncI, IncA/C, IncL, IncN, IncH, and IncX (3). IncF plasmid was discovered by Esther Lederberg

Address correspondence to Luchao Lv, chaolulv@163.com, or Jian-Hua Liu, jhliu@scau.edu.cn.

The authors declare no conflict of interest.

See the funding table on p. 15.

in the 1950s (4), and it was the earliest identified plasmid type. IncF plasmid has become the dominant plasmid type in *Enterobacterales* (5), among which IncF33 plasmid is one of the most frequently encountered types (6) and is also a newly emerged plasmid type in recent years. Previous studies suggest that IncF33 plasmid is mainly prevalent in China, and it has been identified in various *Enterobacterales*, including *Escherichia coli*, *Klebsiella pneumoniae*, *Salmonella*, and *Proteus mirabilis*. IncF33 plasmid-positive isolates are also widely distributed among humans, animals, food, and the environment; however, they are mainly prevalent in China animal-source *E. coli* and are the primary vehicles for transmitting clinically important resistance genes, such as $bla_{CTX-M-55}$, $bla_{CTX-M-65}$, $rmtB$, $fosA3$, and $oqxAB$ (7–15).

pHN7A8, carried by an *E. coli* strain of canine origin collected in 2008 from China, was the first completely sequenced IncF33 plasmid. Its backbone structure was similar to the conventional IncFII plasmid R100 and carried $bla_{CTX-M-65}$, $bla_{TEM-1B}$, $rmtB$, and $fosA3$ genes (16). Notably, following the discovery of pHN7A8, the detection rate of this type of plasmid has been rising in China but only sporadically reported abroad. Interestingly, the IncF33 plasmids p397Kp and p477Kp carried by two clinical *K. pneumoniae* isolates recovered from Bolivia in 2013 were found to be highly related to the China epidemic plasmid pHN7A8 (17). IncF33 plasmids have also been identified in *Enterobacterales* from chickens in Brazil (18), natural settings in the United States (19), cattle in Canada (20), and humans in Egypt (21), indicating that there has been a tendency of international dissemination of IncF33 plasmids. In addition, the recombination of IncF33 plasmid with other plasmids types, such as IncR, IncN, IncFI, IncX1, rolling circle plasmids, and phage-like plasmids, to form novel cointegrate plasmids has been increasingly reported (14, 22–24), and maybe promote the further diffusion of this plasmid.

However, we have limited knowledge of where the IncF33 plasmid first appeared, how it formed, and how did it come to evolve afterward. In this study, we statistically analyzed a global collection of the whole-genome sequences of 863 strains bearing IncF33 plasmid replicons from various origins. We also perform comparative genomics analysis on IncF33 plasmids obtained from in-house data sets and the GenBank nucleotide database to explore the origin and evolution of this plasmid.

## RESULTS AND DISCUSSION

### IncF33 plasmids are widely distributed and are mainly prevalent in China animal-source *E. coli*

We compiled a data set of whole-genome sequences of 3,382 strains carrying IncF33 plasmid replicon and 292 completely sequenced plasmids positive for IncF33 plasmid replicon. The 292 plasmids can be roughly classified into three types based on their backbones: (i) plasmids with typical IncF33 plasmid backbones, some of which have fragments of other plasmid types inserted into variable regions; (ii) cointegrated plasmids formed by the fusion of a complete IncF33 plasmid and a complete plasmid of another type; (iii) complete $bla_{KPC}$-positive IncR plasmids cointegrated with an IncF33 plasmid fragment that lacked most conjugation-associated genes (this type of plasmids was excluded from this study due to the reasons explained in Materials and Methods). Subsequently, we performed a statistical analysis of the isolation source, geographical location, year of isolation, and bacterial host of 103 IncF33 plasmids and 863 strains carrying IncF33 plasmid replicons. The results showed that IncF33 plasmid replicons were identified in 41 countries, including those in Asia (China, Korea, Japan, Laos, Vietnam, Singapore, India, Lebanon, Pakistan, Republic of Bangladesh, Qatar, Myanmar, Russia, and Thailand), Africa (Egypt, Kenya, Uganda, and South Africa), South America (Ecuador, Peru, Bolivia, Brazil, and Chile), North America (USA, Canada, Mexico, Barbados, and Dominican Republic), Europe (Sweden, Estonia, Slovakia, Germany, Switzerland, UK, Denmark, Netherlands, France, Spain, and Italy), and Oceania (Australia and New Zealand). Of note, both IncF33 plasmids (90.29%, 93/103) and whole-genome sequences of strains (58.16%, 481/827) positive for IncF33 plasmid replicons are mainly distributed in China (Fig. 1).

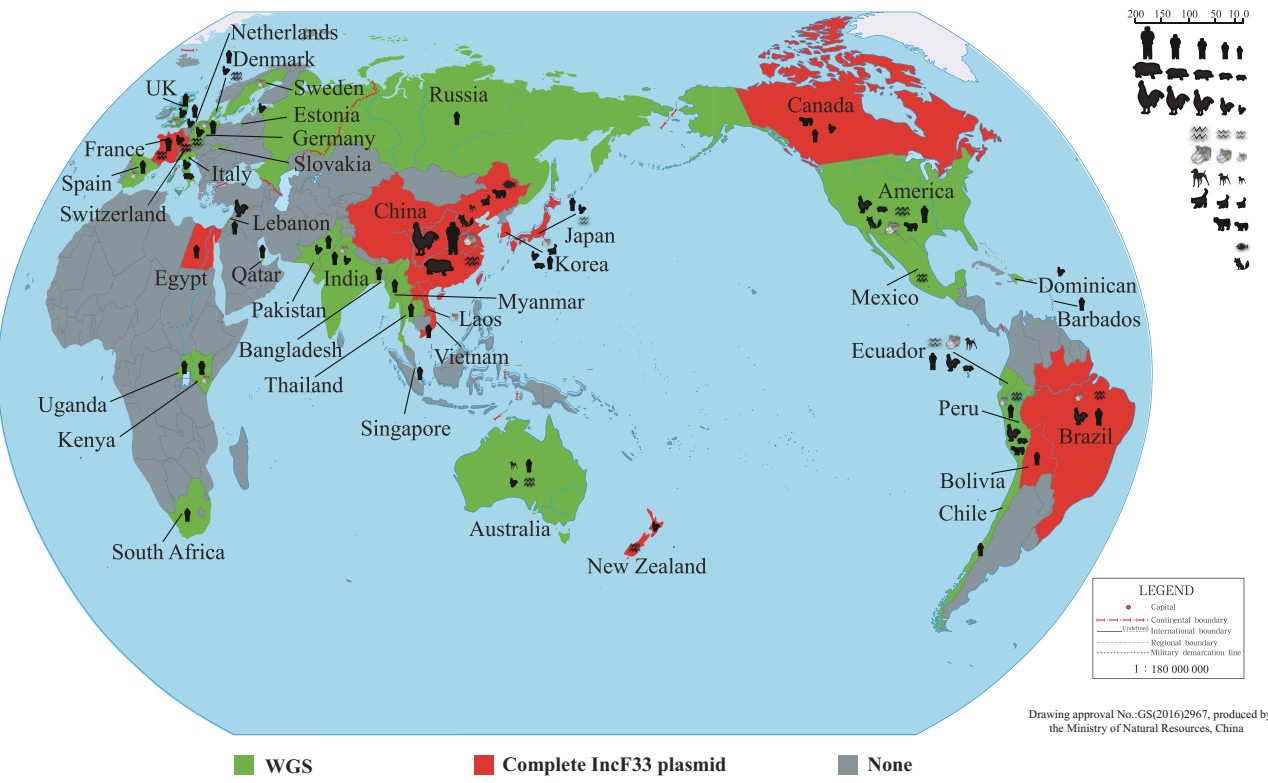

**FIG 1** Global distribution of IncF33 plasmids. Red indicates IncF33 plasmids with fully sequenced genomes. Green indicates whole-genome sequences of IncF33 plasmid replicon-positive strains. The size of the animal silhouette is proportional to the number of strains.

Further statistical analysis of the host of IncF33 plasmids revealed that IncF33 plasmids have a broad host range within the *Enterobacterales*. *E. coli* strains (*n* = 746, 86.44%) are the dominant host, followed by *Salmonella* (*n* = 74, 8.57%) and *K. pneumoniae* (*n* = 30, 3.48%). IncF33 plasmids are also sporadically identified in other *Enterobacterales*, such as *Escherichia fergusonii*, *Escherichia albertii*, *Citrobacter freundii*, *Enterobacter cloacae*, *Enterobacter hormaechei*, *Klebsiella variicola*, *Serratia marcescens*, and *Shigella flexneri*. Furthermore, IncF33 plasmids were identified in *Enterobacterales* isolates from various origins, including animals, humans, food, and the environment. Except for 83 strains for which no source information was found, of the remaining 780 strains, the most common source was animals (58.21%, 454/780), followed distantly by humans (26.03%, 203/780), food (9.36%, 73/780), and the environment (6.67%, 52/780). Animal isolates were predominantly from poultry (48.68%, 221/454) and swine (39.65%, 180/454) (Fig. 2a). These results suggested that IncF33 plasmid is mainly prevalent in China animals, especially in *E. coli* from poultry, and exhibits an increasingly broad distribution trend.

## Multi-locus sequence type diversity in *E. coli*, *K. pneumoniae*, and *Salmonella* carrying IncF33 replicon

Multi-locus sequence typing assigned 669 *E. coli* isolates to 157 distinct sequence types (STs) except for 77 isolates with novel STs, and ST10 (6.43%, 48/746) was most frequently detected (Fig. 2b). In contrast, *K. pneumoniae* and *Salmonella* isolates exhibited dominance of specific STs and sources. *Salmonella* ST11 (54.05%, 40/74) and *K. pneumoniae* ST11 (56.67%, 17/30) were the most frequently detected (Fig. 2c and d). Among the *Salmonella* ST11 and *K. pneumoniae* ST11 isolates, the most common source was humans, accounting for 65% (26/40) and 82.35% (14/17), respectively (Fig. 2e).

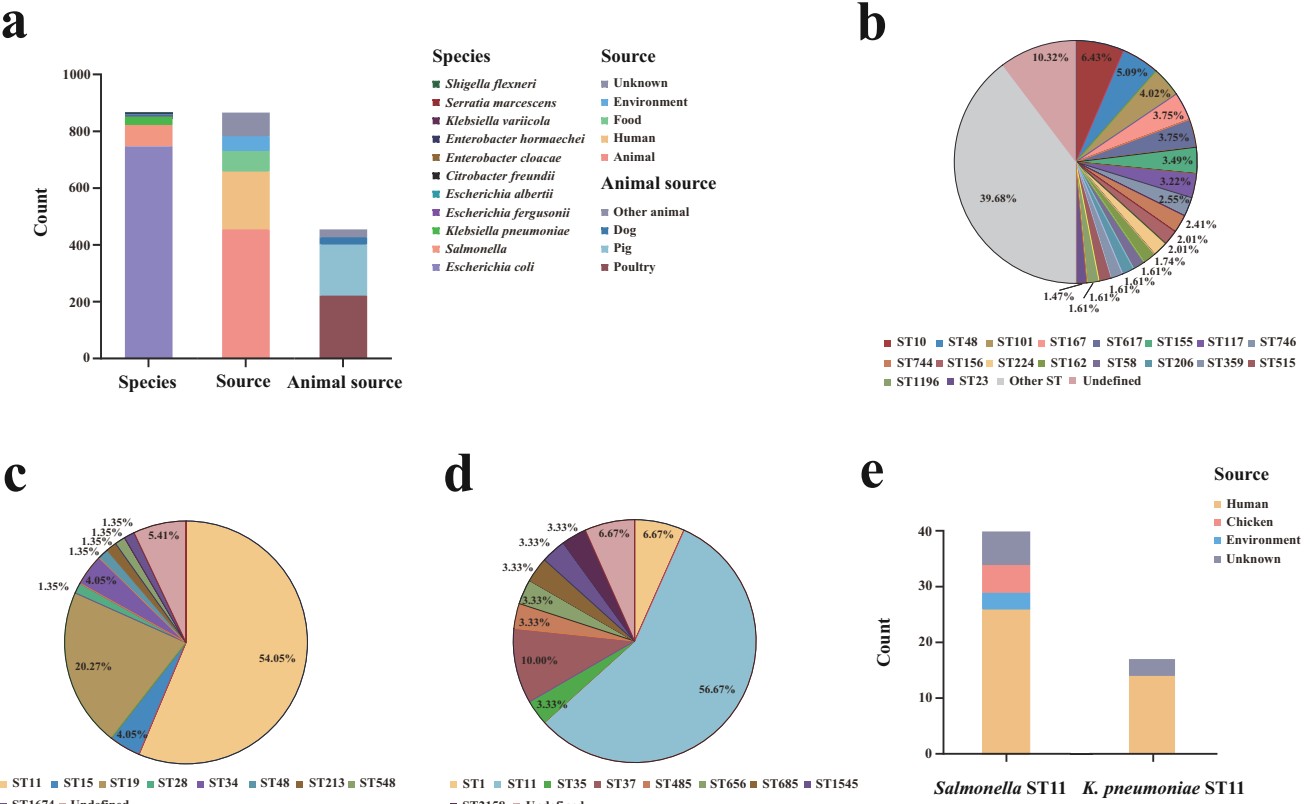

**FIG 2** Distribution of 863 IncF33 plasmid replicon-positive strains by species, source, animal source, and sequence type. (a) Species, source, and animal source distribution of 863 IncF33 plasmid replicon-positive strains. (b) Sequence type of *E. coli* isolates. (c) Sequence type of *Salmonella* isolates. (d) Sequence type of *K. pneumoniae* isolates. (e) Source distribution of *Salmonella* ST11 and *K. pneumoniae* ST11 isolates.

## IncF33 plasmids carry several clinically important resistance genes

The size of the 103 IncF33 plasmids ranged from 39,248 to 284,309 bp. They carried 0 to 15 ARGs, such as beta-lactam resistance genes $bla_{CTX-M-55}$ ($n = 78$, 75.73%) and $bla_{CTX-M-65}$ ($n = 14$, 13.59%), fosfomycin resistance gene *fosA3* ($n = 63$, 61.17%), aminoglycoside resistance genes *rmtB* ($n = 45$, 43.69%), *aph(3″)-Ib* ($n = 31$, 30.10%), *aph(6)-Id* ($n = 31$, 30.10%), and *aph(3′)-IIa* ($n = 25$, 24.27%), amphenicols resistance gene *floR* ($n = 33$, 32.04%), quinolone resistance gene *oqxAB* ($n = 19$, 18.45%), tetracycline resistance gene *tet*(A) ($n = 31$, 30.10%), and sulfonamide resistance gene *sul2* ($n = 31$, 30.10%) (Fig. S1a). Furthermore, $bla_{NDM}$, *mcr-1.1*, *cmlA1*, *qnrS1*, and other ARGs were also identified. Over half of the IncF33 plasmids were found to have more than three resistance genes (Fig. S1b)

## The IncF33 plasmid backbone sequence is highly conserved, and individual plasmids have varying degrees of backbone sequence deletion

To investigate evolutionary relationships within our collection of IncF33 plasmid sequences, phylogenetic analyses were constructed using the core genes of 103 IncF33 plasmids. Each branch was characterized by source, years, country, the number of ARGs, and the presence of group II intron and ARGs. Phylogenetic analysis showed that except for 16 phylogenetically distant plasmids due to the lack of some backbone fragments, most of the plasmids were clustered on the same branch, indicating that the phylogenetic relationship between these plasmids was closer (Fig. 3). Additionally, the results of the linear comparisons of the IncF33 plasmid backbones were consistent with the phylogenetic tree. Most of these plasmids (74.76%, 77/103) have the same backbone

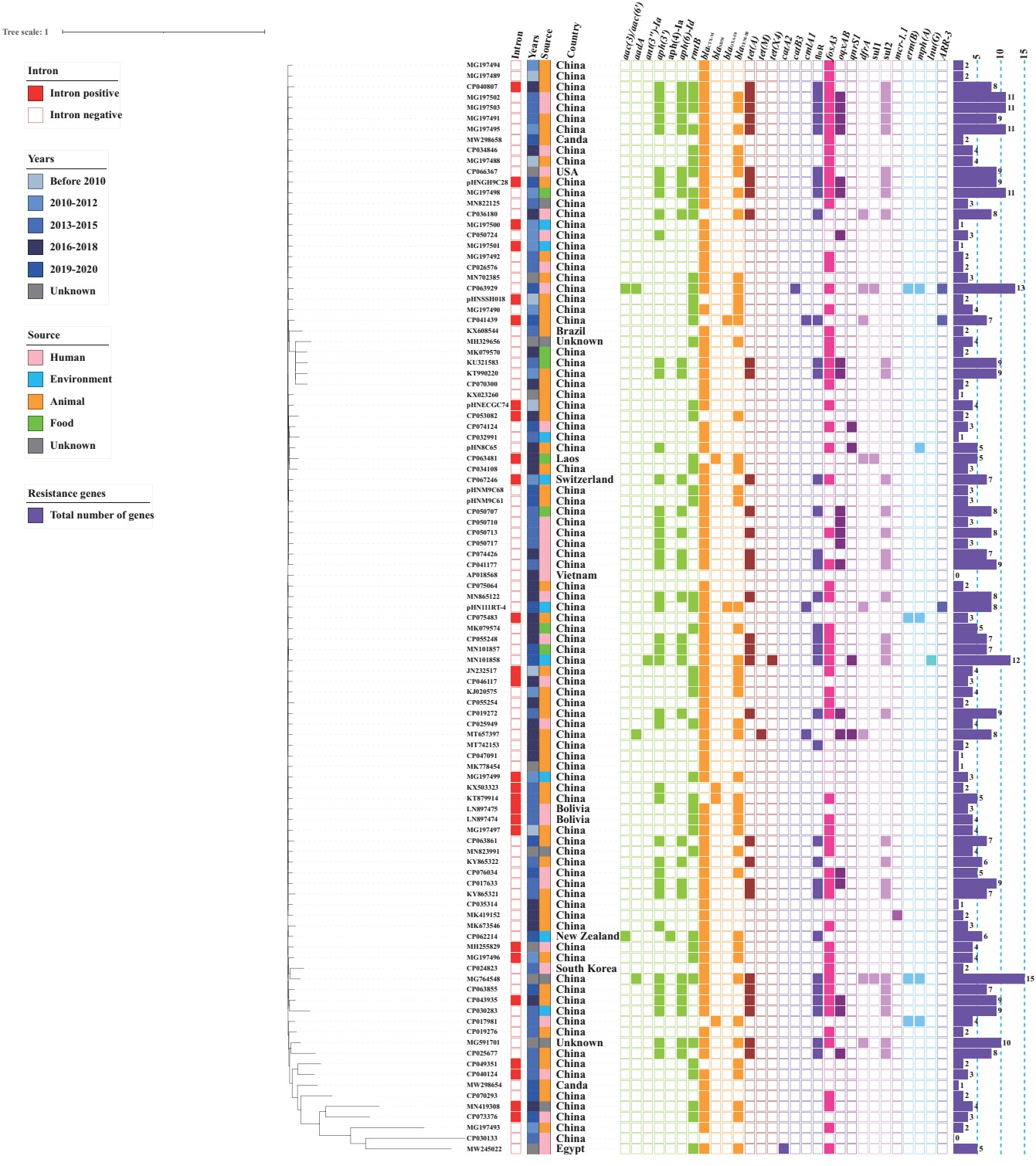

**FIG 3** Maximum-likelihood core genome phylogeny of 103 IncF33 plasmids.

structure as pHN7A8, including replication, leading and transfer regions, and showed over 99% homology (Fig. 4). IncF33 plasmids have the typical structure of IncFII plasmids, containing three major regions: replication region contained genes (*repA2/copB-repA1-repA4*) that control plasmid replication; leading region contained genes (*pemI/pemK*, *psiB/psiA*, *stbB/stbA*, *parB*, *ssb*, *sok/hok/mok*, etc.) related to plasmid maintenance and

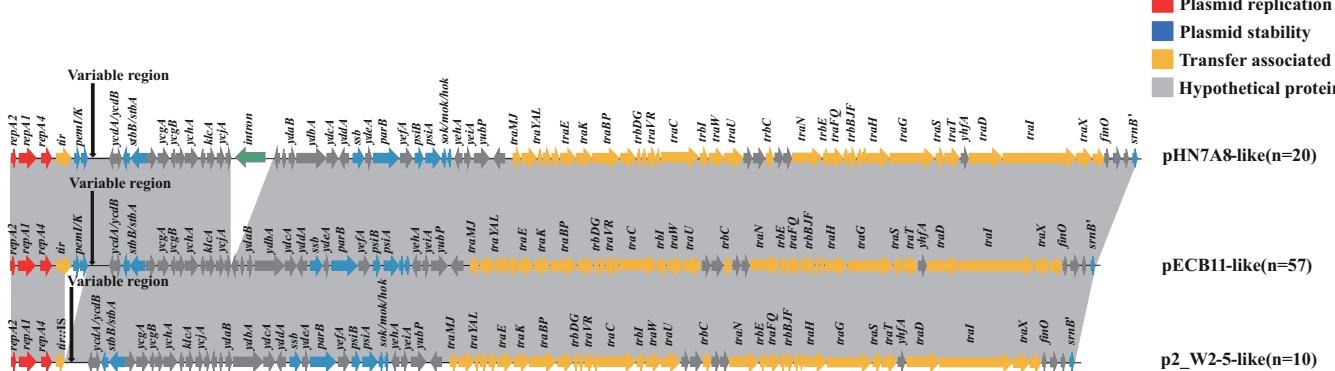

**FIG 4** Linear comparison of IncF33 plasmid backbones. Over 99% homologous segments are indicated by gray shading. Genes are represented by arrows and are classified by function into different groups.

stability; conjugative transfer region contained genes (*traA* to *traY*, *trbB* to *traJ*, *finO*, *srnB'*, etc.) responsible for pilus synthesis and assembly, DNA transfer, aggregate stability, surface exclusion, and regulation of transfer region gene expression (25). The variable region of IncF33 plasmids was located downstream of *pemI/pemK* and contained resistance modules, and the variable region of some IncF33 plasmids also contained backbone segments from other plasmid types. Different degrees of deletion events were found in the other 26 plasmids, such as p2_W2-5-like, p00004, p12478-rmtB, etc. In some incomplete IncF33 plasmid backbone sequences, we observed that the *tir* gene is a hot spot for insertion by IS*1294* or IS*Ec62*. However, these insertion events typically result in the deletion of the addiction system *pemI/K*, which was located downstream of *tir*. Moreover, partial leading genes (*ssb-ydeA-parB-yefA-psiBA-sok/hok/mok*) and transfer region genes (*traW* to *traI*) were also frequently deleted, possibly due to a recombination event. The leading genes and partial transfer regions genes of pLSH-KPN148-2 were inverted compared with those of pHN7A8. Similarly, pHNAH9 and pIncFII_L111 lacked a large part of leading and transfer regions, which might also result from multiple recombination events (Fig. S2 to S4). These deletion events were not associated with other factors like bacterial host, isolation source, geographical location, year of isolation, etc.

## IncF33 plasmid has the potential for intercontinental dissemination

Comparative analysis of IncF33 plasmid backbones in different countries showed that the p397Kp (LN897474) and p477Kp (LN897475) plasmids from Bolivia were highly similar to China epidemic plasmid pHN7A8 (JN232517), and there were only nine single nucleotide polymorphisms (SNPs) between p397Kp and pHN7A8 (17). Of note, pT224A (MW298658) from bovine in Canada, unnamed plasmid (CP066367) from humans in the United States, and unnamed plasmid (CP062214) from water in New Zealand were all highly similar to the pHN7A8 plasmid backbone (Fig. 5). We hypothesized that these plasmids are closely related to pHN7A8. To test this, we determined pairwise SNP counts between these plasmid sequences (Fig. 6). The analysis showed less than 15 SNPs between these plasmids, among which the plasmid MW298658 differed from the plasmid CP062214 by only two SNPs. The above results indicated that IncF33 plasmid might be transmitted across regions in China, Bolivia, the United States, Canada, New Zealand, and other countries, highlighting their potential for intercontinental dissemination.

## The ancestor of the IncF33 plasmids

To investigate the evolutionary relationship within our collection of IncF33 plasmids, we aligned the whole-genome sequences of strains positive for IncF33 plasmid replicons

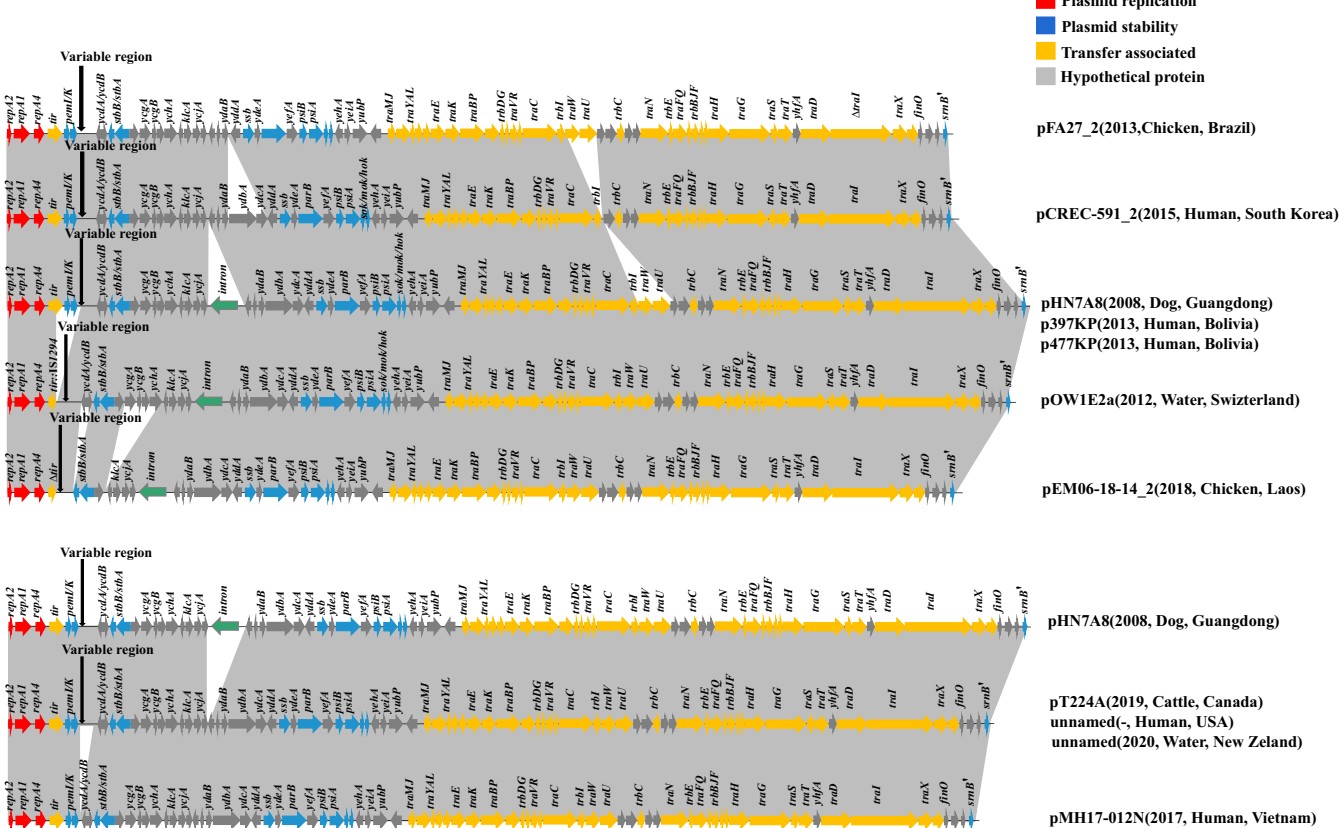

**FIG 5** Linear comparison of IncF33 plasmid backbones in different countries.

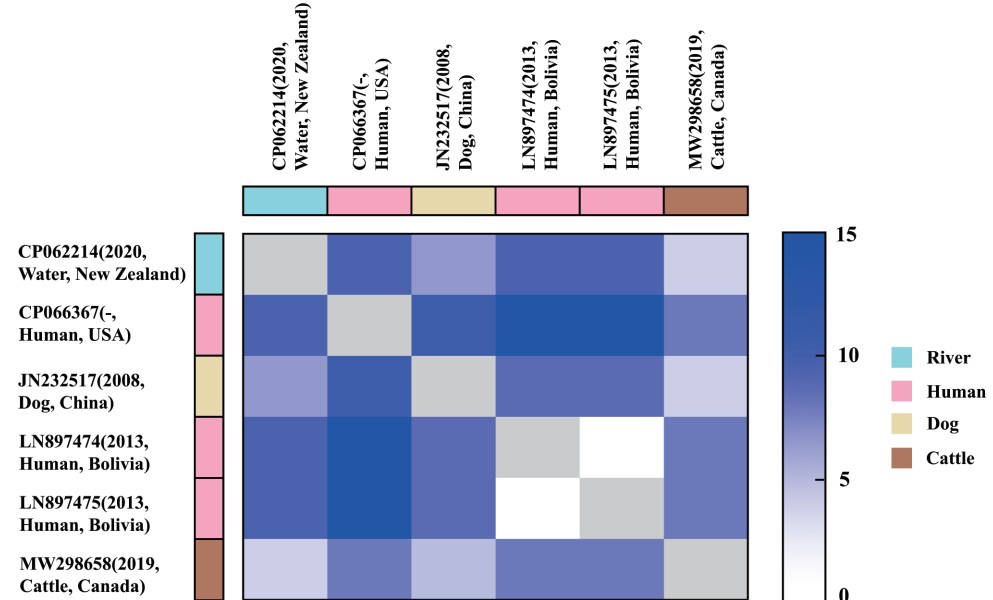

**FIG 6** Close genetic relationships between IncF33 plasmids from different countries. Color scale from white to dark blue indicates increasing numbers of single nucleotide polymorphisms.

to the backbone of pHN7A8. We spliced the matched contigs to obtain the complete plasmid, and four early IncF33 plasmid sequences were obtained from their

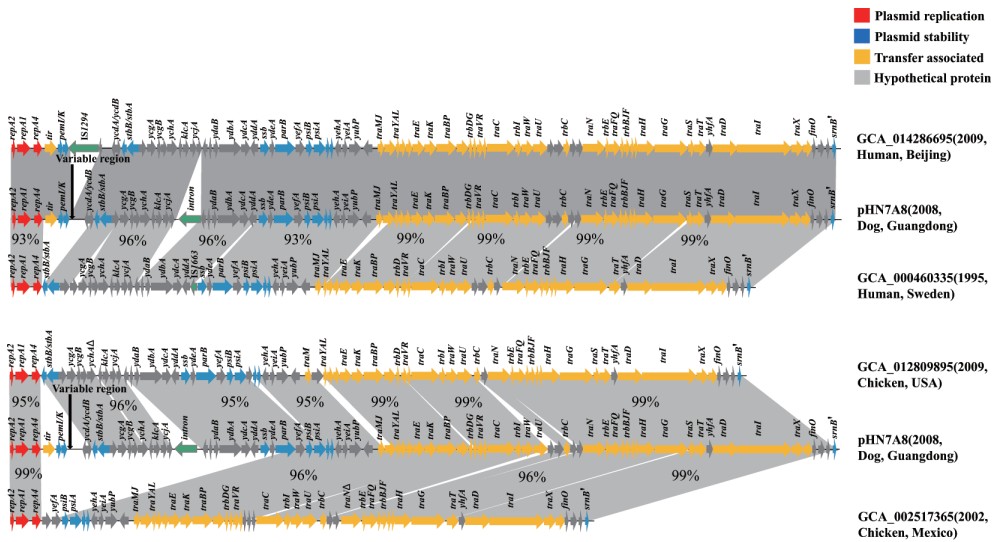

**FIG 7** Linear comparison of the earlier dated IncF33 plasmid backbones and pHN7A8 (IncF33) backbone.

hosts, i.e., GenBank accession numbers GCA_000460335.1 (1995, Human, Sweden), GCA_002517365.1 (2002, Chicken meat, Mexico), GCA_012809895.1 (2009, Chicken meat, USA), and GCA_014286695.1 (2009, Human, China). For convenience, the GenBank accession numbers of the strains are tentatively referred as their plasmids in subsequent analysis. Linear comparison of four plasmids with pHN7A8 plasmid backbone showed that the group II intron was not detected in all four plasmids, and plasmid GCA_014286695.1 possessed the same backbone as plasmid pHN7A8 except for *pemK* which was truncated by IS*1294*. In addition, no ARGs were identified in plasmid GCA_014286695.1. The other three plasmids were all missing an approximately 3.6 kb segment containing the *tir*, *pemI/K*, *ycdA*, and *ycdB* genes. Plasmid GCA_002517365.1 also lost most of the leading region genes and putative genes downstream of this segment, including *stbB/A*, *ycgA/B*, *ssb*, and *parB*, etc. (Fig. 7). Consequently, we hypothesize whether the *tir-pemI/K-ycdA-ycdB* segment was obtained subsequently. To test this, we retrieved the *tir-pemI/K-ycdA-ycdB* segment and searched plasmids positive for this segment in the National Center for Biotechnology Information (NCBI) database. We found that *tir-pemI/K-ycdA-ycdB* was presented in F2:A-:B- plasmids (CP038385 and CP025139) from *E. coli* isolates from humans in Iran in 1963 and *E. coli* isolates from humans in Brazil in 1974, respectively. Linear comparisons of these two plasmids with pHN7A8 plasmid backbone showed that the *tir-pemI/K-ycdA/B-stbB/A-ycgA/B-ychA-klcA-ycjA* segment showed 99% identity to the corresponding segment of pHN7A8, but the entire conjugative transfer region genes differ (88% coverage and 96% identity) from those of pHN7A8, with lower homology to the corresponding genes of pHN7A8, such as *traM*, *traJ*, *trbD*, *traN*, *trbA*, and *traS* genes. These findings suggested that early IncF33 plasmids, such as GCA_000460335.1, GCA_002517365.1, and GCA_012809895.1, may have acquired the *tir-pemI/K-ycdA-ycdB* segment or a larger segment of the leading region from the F2:A-:B- plasmids, giving them a more complete leading region that facilitates their efficient persistence among *Enterobacterales* and widespread dissemination in the environment, and eventually evolved into the more prevalent IncF33 plasmid lineage (Fig. 8).

## The variable regions of IncF33 plasmids are highly variable and often contain fragments of other plasmid types

Clustering analysis of the variable region sequences of 103 IncF33 plasmids using CD-HIT program (version 4.8.1) showed that the variable region sequences of all plasmids were assigned to 65 clusters, of which 59 clusters were represented by single or two plasmids.

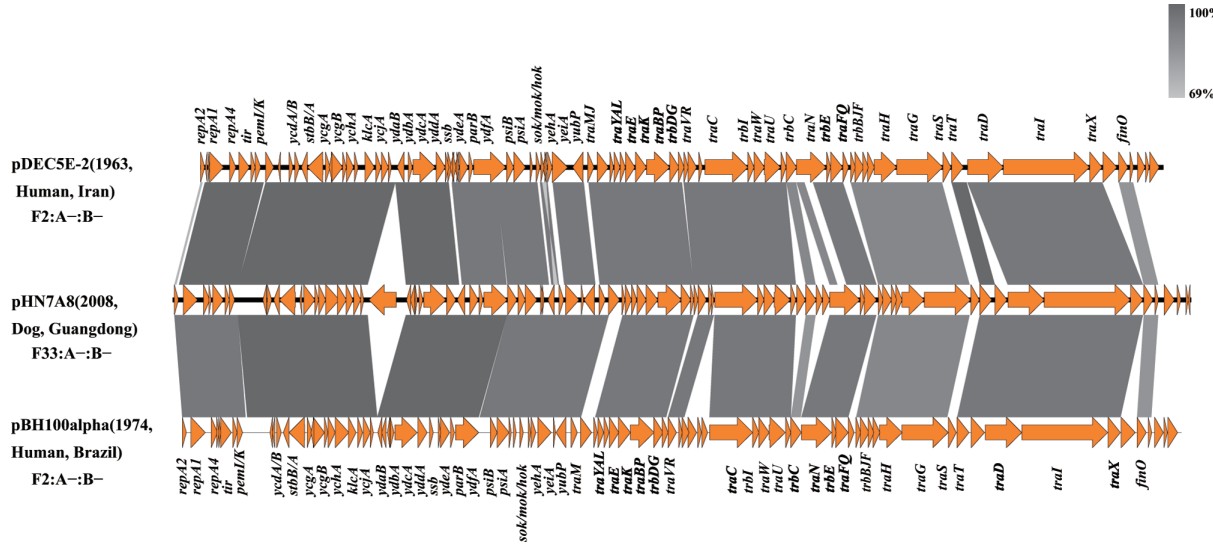

**FIG 8** Linear comparison of the F2:A-:B- plasmid backbones and pHN7A8 (IncF33) backbone.

The remaining six clusters comprised three or more plasmids. Cluster A contained the largest number of plasmids (*n* = 8), and several other major clusters included clusters B (*n* = 4), C (*n* = 3), D (*n* = 4), E (*n* = 4), and F (*n* = 3). The clustering results showed that the variable regions of IncF33 plasmids are highly diverse.

### Variable region of cluster A plasmids

Plasmids pAH01-3, pT224A, pHNMC02, pHNBC-6, and pHNGD4P177 possessed variable regions similar to that of F2:A-:B- plasmid pHK23a (JQ432559) from a porcine-derived *E. coli* isolate in China in 2008 and F2:A-:B- plasmid CP020340 from a human-derived *Shigella flexneri* isolate in China in 2007 (Fig. 9), but the *aac(6')-lb-cr* resistance module [IS*26*-hp-*aac(6')-lb-cr*-ΔIS*26*-IS*26*] in CP020340 was not found in these IncF33 plasmids. This observation may be associated with the recombination between the two copies of IS*26* in the same orientation that could lead to the insertion or loss of the resistance module. At the same time, a segment found in these IncF33 plasmids and CP020340, containing three IS*26* elements, truncated *intl1*, and the *fosA3* resistance module, was identical to that of pHK23a with opposite orientation. In addition, *bla*CTX-M-3 was found in plasmid pHK23a and CP020340 with two SNPs compared to *bla*CTX-M-55 in those IncF33 plasmids.

Plasmids pCREC-591_2, pHNMPC43, and pHNMPC51 possessed similar variable regions to pAH01-3, but a segment (*intl1Δ*-IS*26*-*orf3Δ*-*orf2*-*orf1*) was absent in pCREC-591_2, which might be the result of a recombination event between *intl1Δ* and *orf1*. In addition, plasmids pHNMPC43 and pHNMPC51 lacked the *fosA3* module and one IS*26* element compared with pAH01-3, possibly due to the recombination events between IS*26* elements.

### Variable region of cluster B plasmids

Compared with cluster A plasmids, the variable region of cluster B plasmids acquired segments from other types of plasmids and became more complex. Briefly, it was divided into six regions by IS*26* or IS*1294* (Fig. 10). The first segment (ΔIS*1*-ΔTn*2*-IS*26*-*intl1Δ*) was identical to segments in the cluster A plasmids with a size of 2.8 kb.

The second segment (~15 kb) consisted of IS*CR2* elements, incomplete transposon Tn*AO22*, and resistance genes such as *floR*, *ΔtetA/tetR*, *strA/ΔstrB*, and *sul2*. This segment showed 99% identity to IncA/C plasmid lacking IS*5075*-Tn*A022* regions such as

## Cluster A

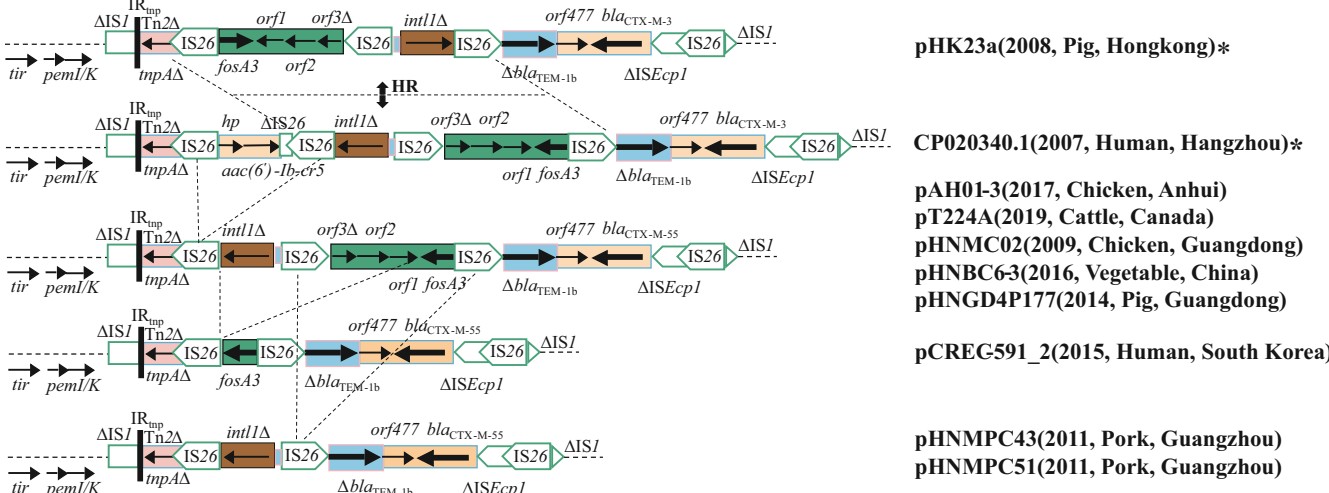

**FIG 9** Comparison of variable regions in cluster A plasmids. The plasmid backbones are represented by dotted lines. Antibiotic resistance genes and other genes are indicated by thick arrows and thin arrows, respectively. Insertion sequences are denoted by boxes labeled with the insertion sequence (IS) name. Arrows marked with "HR" and dotted lines signify that differences between structures may have resulted from homologous recombination. Dotted diagonal lines indicate possible deletion and insertion events. F2:A-:B- plasmids are marked with "*".

## Cluster B

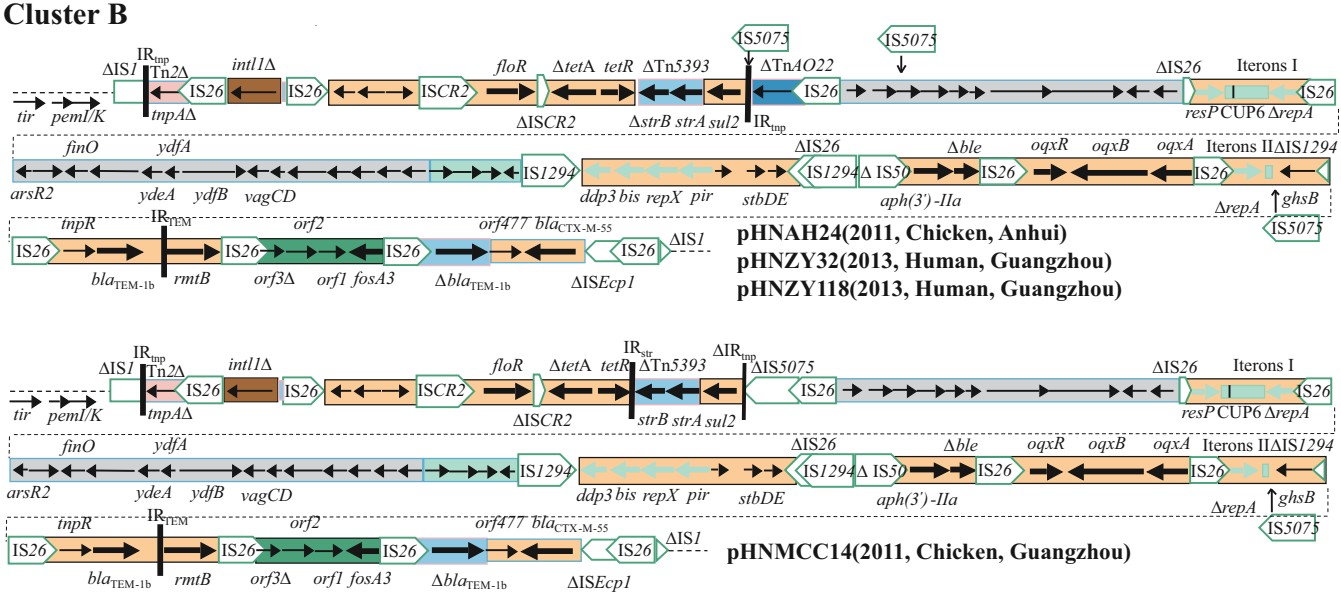

**FIG 10** Comparison of variable regions in cluster B plasmids. Adjacent regions are connected with dotted lines.

pSH11G0791 (CP041172.1, *Salmonella enterica*) and pST56-1 (CP050740.1, *Salmonella enterica*).

The third segment contained two parts. The first part (~15 kb) was bounded by two copies of IS*26* in the same orientation and included 10 putative open reading frames (ORFs) and an IncN1 pABWA45_3-like plasmid segment (*resP*-CUP6-Iterons I-*ΔrepA*). The second part (~18 kb) was bounded by IS*26* and IS*1294* in the opposite orientation. It contained two regions; the first region (~15 kb) was highly homologous to that of IncI1 plasmid pCombat13F7-3 (CP019248), which was carried by an *E. coli* strain of human

origin in 2004 from China. The second region (~2.9 kb) contained four putative ORFs and was found in F-:A13:B- plasmids, such as pK245 (2002, DQ449578.1), pKp2177_2 (2008, CP075593.1), and p08EU827_2 (2008, CP025578.1). Therefore, this ~18 kb hybrid part might have been generated, owing to the recombination between an IncFIA-like fragment and an IncI1 fragment.

The fourth segment (*ddp3-bis-repX-pir-stbD/E*) showed 98% identity to IncX1 plasmid pOLA52 (EU370913) lacking *repX* gene. The generation of this segment might involve the insertion and transposition of multiple IS elements, including IS*26*, IS*1294*, and IS*50*. The fifth segment contained *aph(3')-IIa*, *Δble*, and *oqxAB*. The *oqxAB* was embedded in composite transposon Tn*6010* (IS*26-oqxA-oqxB-oqxR*-IS*26*). The last segment (~10 kb) was identical to the segments of the cluster A plasmids and contained $bla_{TEM-1b}$, *rmtB*, *fosA3* resistance module, and $bla_{CTX-M-55}$.

In addition, compared with pHNAH24, pHNZY32, and pHNZY118, the variable region of plasmid pHNMCC14 lacked 14 bp of the 5' end of IS*5075*, incomplete Tn*AO22*, and one complete IS*5075* element. In summary, cluster B plasmids acquired more ARGs and fragments from other types of plasmids, such as IncN, IncX1, InFIA, and IncI1.

### Variable region of cluster C plasmids

Plasmids pHNFKD271 and pHN04NHB3 possessed a similar organization as the third segment (IncN, IncFIA, and IncI1 hybrid segment) of cluster B plasmids but in an opposite orientation. The variable regions of pHNFKD271 and pHN04NHB3 may have been generated from pHK23a-like (cluster A-like plasmid) multidrug resistance region (MRR) by IS*1294*-mediated hybrid segment insertion plus IS*26*-mediated homologous recombination (Fig. 11).

Compared with pHNFKD271 and pHN04NHB3, the IncN replication region was not found in plasmid p103-2-4, probably due to the IS*26*-mediated homologous recombination.

### Variable region of cluster D plasmids

Plasmids pECB11 and p2019XSD11-92 had variable regions similar to cluster B plasmids but with the deletion of several fragments. The first fragment contained incomplete *intI1*, three putative ORFs, and two IS*26* elements. The second fragment was an IncN segment without IS*5075* insertion but with an opposite orientation. The third fragment contained the IncN replication region, IncI1 segment, IncFIA-like segment, IncX1 segment, composite transposon Tn*6010*, and the second IncN1 segment. In addition, the variable regions of pZY-1 and pM1023-4Ar.4 were related to pECB11 but with the deletion of different quantities of putative ORFs in the IncN segment (Fig. 11).

### Variable regions of cluster E and cluster F plasmids

The variable regions of cluster E plasmids differed from those of clusters A, B, C, and D plasmids, containing the typical transposition unit (IS*Ecp*1-$bla_{CTX-M-65}$-IS*903-iroN*), the *fosA3* module, and the resistance region bounded by IS*1294* or IS*26*. BLASTn analysis indicated the high similarity (95% coverage and 99% identity) of pHN7A8 MRR to plasmid ps12177-CTX (CP101349) recovered from human origin in Henan Province, China, in 2006 (Fig. 12).

Cluster F plasmids possessed variable regions similar to cluster E plasmids and differed from cluster E plasmids by deletion of the *fosA3* module, maybe resulting from a recombination event between the two copies IS*26* in the same orientation (Fig. 12).

### Variable regions of remaining plasmid clusters

Due to current database limitations, the remaining 59 clusters were represented by single or two plasmids, and the structures of their variable regions are mostly

## Cluster C

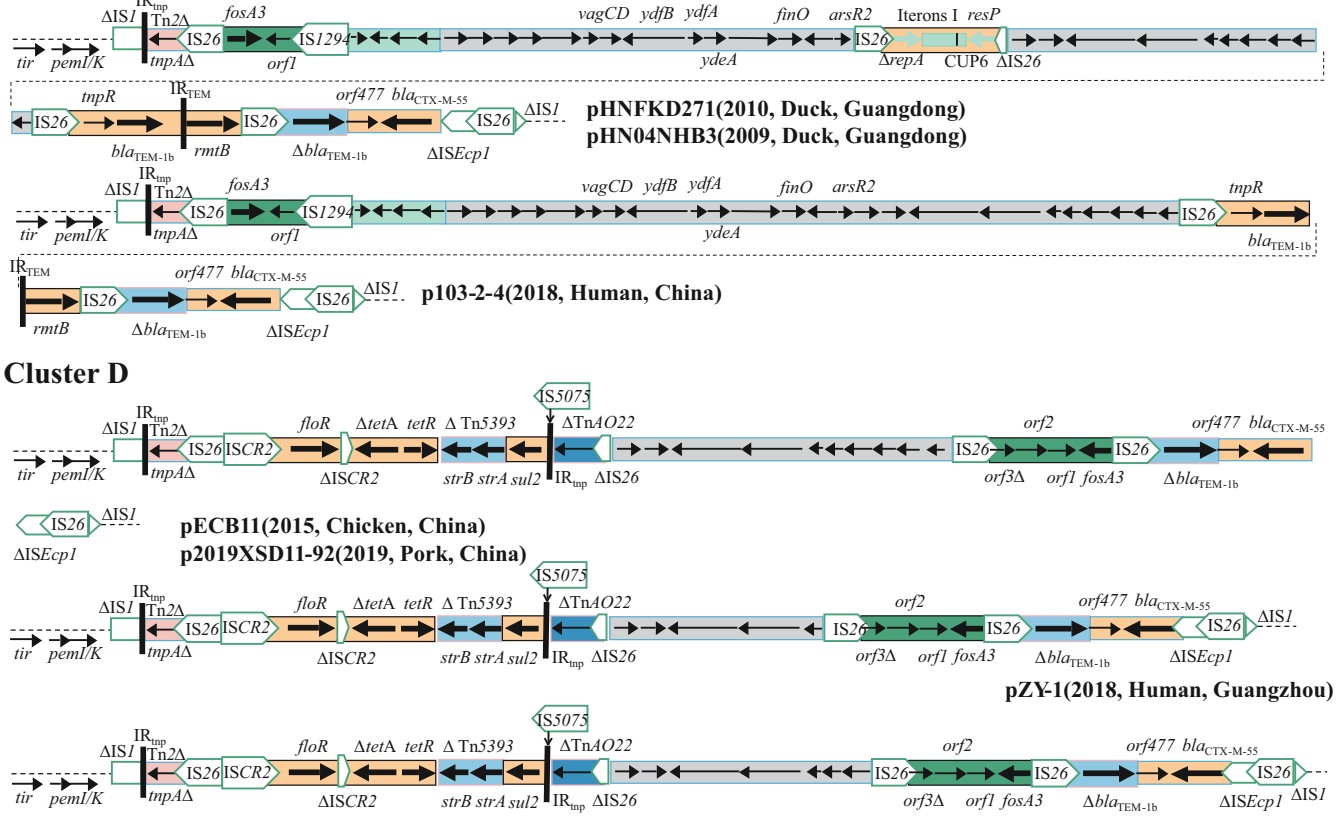

**FIG 11** Comparison of variable regions in clusters C and D plasmids. Adjacent regions are connected with dotted lines.

## Cluster E

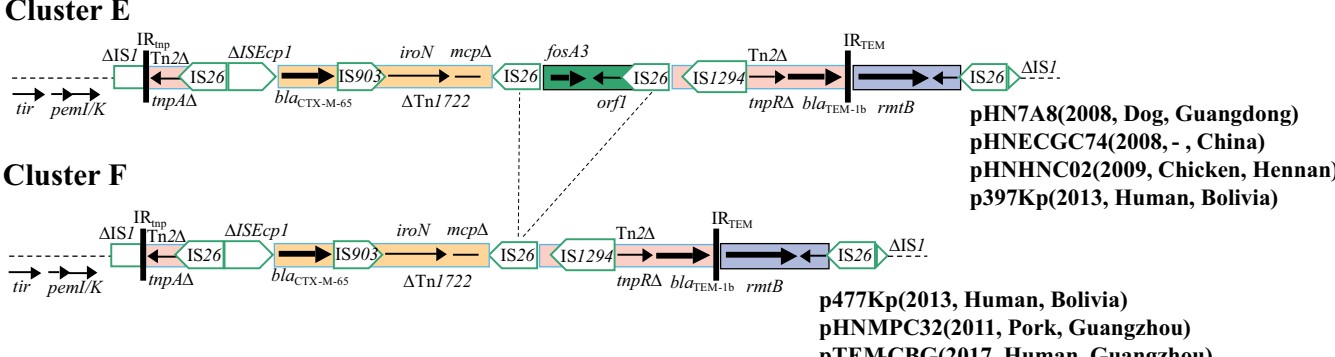

**FIG 12** Comparison of variable regions in clusters E and F plasmids. Dotted diagonal lines indicate possible deletion and insertion events.

intermediate or transient forms in the evolution of major variable region types. They deserve attention in future studies with larger plasmid data sets.

To summarize, all IncF33 plasmids contain the same IS*1*/backbone boundary as the prototype R100 plasmid, indicating that the variable region of the IncF33 plasmid may have evolved from the variable region of R100 plasmid through a series of steps. After long-term evolution, the variable regions of IncF33 plasmids have become highly diverse, which provide multiple extra functions that enable their hosts to adapt to changing environmental conditions and increase the adaptability of the host bacteria.

Currently, the variable regions of IncF33 plasmids can be roughly divided into two sublines based on the production of different CTX-M enzymes. The first subline includes cluster_ABCD-like plasmids producing CTX-M-55, which have evolved from cluster_A through some insertion (fragments of other types of plasmids) steps. The second subline includes cluster_EF-like plasmids producing CTX-M-65, which have a significantly different structure. Moreover, IncF33 plasmids carrying the variable region producing CTX-M-55 are more dominant, and their sources are more diverse. On the other hand, IncF33 plasmids carrying the variable region producing CTX-M-65 are more frequently found in human and chicken sources. Interestingly, only IncF33 plasmids carrying $bla_{\text{CTX-M-55}}$ have been identified in pig and environmental sources, while $bla_{\text{CTX-M-65}}$ has not been found. This may be related to the fact that cluster_ABCD-like variable regions carry more insertion fragments, including replicon fragments of other types of plasmids, which may facilitate the spread of plasmids carrying this type of variable region across different hosts (26, 27).

## The cointegration of IncF33 plasmids with other types of plasmids becomes more common

In recent years, the IncF33 plasmids have generated cointegrates with various plasmids, among which, we observed two different types of cointegrates. One was formed by insertion of fragments from other plasmid types into the variable regions of complete IncF33 plasmids (type I). The other resulted from fusion of complete IncF33 plasmids with plasmids of other types, mediated by homologous recombination involving IS elements or transposons (type II). The former (41.75%, 43/103) was more common than the latter (2.91%, 3/103) within our collection of 103 IncF33 plasmids (excluding $bla_{\text{KPC}}$-positive IncR-IncF33 plasmids).

So far, various types of plasmids, including IncR, IncN, IncFI, IncX1, rolling-circle plasmids, and phage-like plasmids, were fused with IncF33 plasmids, forming the first type of cointegrates. Previous studies confirmed that the truncation of *traI* by the R plasmid insertion event resulted in IncF33:R cointegrate abolishing transfer ability. Still, in some other fusion cases, the cointegrates had relatively high conjugation frequencies and stability, likely stimulating the dissemination of IncF33 plasmids (22, 24). In addition, we used the Plasmidfinder database to identify additional plasmid replicons within our collection of 103 IncF33 plasmids for detecting the second type of cointegrates. Finally, additional plasmid replicons were found in 43 (41.75%) plasmids, including IncN1, IncX1, IncN1-X1, rolling-circle, IncHI1A, and IncHI1B. Of these, the most common additional replicon was IncN1 (27.18%, 28/103), followed by IncX1 (22.33%, 23/103), and IncN1-X1 (8.74%, 9/103), and other replicons were identified only sporadically. Therefore, the cointegration of IncF33 plasmids with other types of plasmids was more common than previously thought.

The multiple ARGs and replicons carried by the cointegrates could expand their resistance and host spectrum, which was conducive to their rapid transmission in *Enterobacterales*, posing a serious threat to public health security.

## Limitations

Studies using public genome databases are often plagued by uncertainty, i.e., whether the entire data set can fully represent the entire population. Sampling bias may exist. Additionally, this study intentionally excluded strains and plasmids carrying $bla_{\text{KPC}}$ (as explained in the methods), analyzing only typical IncF33 plasmids. However, it is known that IncR-IncF33 cointegrate plasmids lack complete conjugative transfer regions and thus cannot spread horizontally. They disseminate primarily via ST11 *K. pneumoniae* clones (23).

## Conclusions

IncF33 plasmid is mainly prevalent in *E. coli* of animal origin in China, particularly in chickens, carries various clinically important ARGs, and is becoming increasingly widespread. Over half of the plasmids carry four or more ARGs. The ancestor of the currently prevalent IncF33 plasmid may not have the *tir-peml/K-ycdA-ycdB* fragment which might have been obtained from early F2:A-:B- plasmids. Additionally, the variable region of the IncF33 plasmid has become increasingly complex through multiple insertion or recombination events and can be broadly divided into two categories. The first category produces CTX-M-55, which carries more ARGs and other plasmid fragments than the second category that produces CTX-M-65, which may be more favorable for its cross-host transmission. Therefore, the acquisition of the *tir-peml/K-ycdA-ycdB* fragment and the continuous evolution of the variable region by insertion or recombination have laid the foundation for the prevalence of the IncF33 plasmid in China and globally. Furthermore, the cointegration phenomenon of the IncF33 plasmid needs continued attention. Cointegrated plasmids carrying multiple ARGs and replication modules can expand their resistance and host spectrum, making them more favorable for rapid transmission among *Enterobacterales*.

## MATERIALS AND METHODS

### Strain and plasmid collection

The sequence of the IncF33 plasmid replicon was downloaded from PubMLST ([https://pubmlst.org/organisms/plasmid-mlst/](https://pubmlst.org/organisms/plasmid-mlst/)) and was used to match the GenBank assembly database. Whole-genome sequences of strains with matches identical to the replicon sequence were downloaded for subsequent analysis. To ensure the reliability of the results, study data were cleaned based on the following methods. (i) The low-quality sequences were excluded from the study. (ii) Since the IncR-IncF33 cointegrated plasmids carrying $bla_{KPC}$ genes were formed by insertion of IncF33 plasmid fragments (lacking most conjugative transfer regions) into IncR plasmid backbones, their backbones remained IncR plasmids rather than typical IncF33 plasmids, which did not align with our research aims. Therefore, the strains and plasmids carrying $bla_{KPC}$ genes were excluded from this study. (iii) Duplicate sequences uploaded by the same institution simultaneously were excluded from this study. Finally, the whole-genome sequences of 863 strains were obtained. Similarly, the IncF33 plasmids were collected from the GenBank nucleotide database and were cleaned using the same methods. Finally, the complete sequences of 292 plasmids were obtained, including 103 plasmids negative for $bla_{KPC}$ genes. In addition, we designed an automated Python program to extract information about the strains and plasmids.

### Sequencing and annotation

Whole genomic DNAs of strain positive for IncF33 plasmid replicons were extracted and were fully sequenced using a HiSeq platform (Illumina, San Diego, CA, USA). Annotation was performed using the RAST server (28), Prokka (29), and ISfinder (30).

### Bioinformatic analysis

The multilocus sequence typing (MLST) were identified using ABRicate 0.9.8 (28). ARGs and plasmid replicons were detected using publicly available ResFinder and Plasmid-Finder databases from the Centre for Genomic Epidemiology (31–34). Core genome of plasmids was inferred using panaroo (35). A phylogenetic tree was inferred from core genome alignment with RAxML (36). Furthermore, the phylogenetic tree was visualized and beautified using online tool iTOL (37).

## Plasmid backbone and variable region

Almost all IncF33 plasmids have obvious insertion sequence (IS*1*, IS*1294*, etc.) boundaries in the variable regions and the locations are all downstream of the *pemI/K* addiction system; thanks to this, we can easily distinguish it in an artificial way. After removing the variable region of the IncF33 plasmid, we use the remaining sequences as the input to the panaroo software, with the --core_threshold parameter set to 0.95, and finally obtain the backbone sequence of the IncF33 plasmid.

## Comparative genomic analysis of plasmids and SNP analyses

The plasmid backbone and variable region sequences were compared using Easyfig (38) and SnapGene (https://www.snapgene.com/). CD-HIT (39) was used for clustering analysis of the plasmid backbone and variable region sequences. SNPs were identified by running Snippy (https://github.com/tseemann/snippy) on the core gene alignment. Pairwise SNPs between plasmids were counted from the core SNP alignment with snp-dists 0.6.3 (https://github.com/tseemann/snp-dists).

## ACKNOWLEDGMENTS

This study was supported by the National Natural Science Foundation of China (No. 32141002 and 31625026), the Local Innovative and Research Teams Project of Guangdong Pearl River Talents Program (2019BT02N054), and the Laboratory of Lingnan Modern Agriculture Project (No. NT2021006).

## AUTHOR AFFILIATIONS

[1]State Key Laboratory for Animal Disease Control and Prevention, Guangdong Laboratory for Lingnan Modern Agriculture, College of Veterinary Medicine, South China Agricultural University, Guangzhou, Guangdong, China
[2]Key Laboratory of Zoonosis of Ministry of Agricultural and Rural Affairs, Guangdong Provincial Key Laboratory of Veterinary Pharmaceutics Development and Safety Evaluation, National Risk Assessment Laboratory for Antimicrobial Resistance of Microorganisms in Animals, Guangzhou, Guangdong, China

## AUTHOR ORCIDs

Guolong Gao  http://orcid.org/0009-0009-2830-3939
Wanyun He  http://orcid.org/0000-0001-8551-9888
Luchao Lv  http://orcid.org/0000-0003-4978-841X
Jian-Hua Liu  http://orcid.org/0000-0002-3930-7857

## FUNDING

| Funder | Grant(s) | Author(s) |
| --- | --- | --- |
| MOST \| National Natural Science Foundation of China (NSFC) | 32141002, 31625026 | Jian-Hua Liu |
| Local Innovative and Research Teams Project of Guangdong Pearl River Talents Program | 2019BT02N054 | Jian-Hua Liu |
| Laboratory of Lingnan Modern Agriculture Project | NT2021006 | Jian-Hua Liu |

## AUTHOR CONTRIBUTIONS

Guolong Gao, Data curation, Formal analysis, Investigation, Methodology, Resources, Software, Validation, Visualization, Writing – original draft, Writing – review and editing | Wanyun He, Methodology, Software, Supervision, Validation, Visualization, Writing – review and editing | Yanxiang Jiao, Data curation, Investigation, Software, Visualization | Zhongpeng Cai, Software, Validation, Visualization | Luchao Lv, Data curation, Formal

analysis, Methodology, Software, Validation, Visualization, Writing – review and editing | Jian-Hua Liu, Conceptualization, Data curation, Formal analysis, Funding acquisition, Investigation, Methodology, Project administration, Resources, Supervision, Validation, Visualization, Writing – review and editing

## DATA AVAILABILITY

The complete sequences of plasmids pHNGH9C28, pHN111RT-4, pHN8C65, pHNECGC74, pHNM9C61, and pHNSSH018 were deposited under the accession numbers CP125894.1, CP125892.1, CP125886.1, CP125888.1, CP125884.1, and CP125890.1.

## ADDITIONAL FILES

The following material is available online.

### Supplemental Material

**Fig. S1 (mSystems00508-23-s0001.eps).** Antibiotic resistance genes carried by IncF33 plasmids.
**Fig. S2 (mSystems00508-23-s0002.eps).** Linear comparison of IncF33 plasmids lacking some backbone fragments.
**Fig. S3 (mSystems00508-23-s0003.eps).** Linear comparison of IncF33 plasmids lacking some backbone fragments.
**Fig. S4 (mSystems00508-23-s0004.eps).** Linear comparisons of IncF33 plasmids lacking some backbone fragments.
**Legends (mSystems00508-23-s0005.docx).** Legends to Fig. S1 to S4.
**Supplemental tables (mSystems00508-23-s0006.xlsx).** Tables S1 to S3.
**Table S4 (mSystems00508-23-s0007.docx).** Information on IncF33 cointegrate plasmids.

### Open Peer Review

**PEER REVIEW HISTORY (review-history.pdf).** An accounting of the reviewer comments and feedback.

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
