## [Reviewer comments · mSystems]

The origin and evolution of IncF33 plasmids based on large-scale datasets

Jian-Hua Liu, luchao Lv, Guolong Gao, Wanyun He, YanXiang Jiao, and Zhongpeng Cai

Corresponding Author(s): Jian-Hua Liu, South China Agricultural University College of Veterinary Medicine

Review Timeline:

Submission Date:	May 22, 2023
Editorial Decision:	June 27, 2023
Revision Received:	July 19, 2023
Accepted:	August 2, 2023

Editor: Robert Beiko

Reviewer(s): Disclosure of reviewer identity is with reference to reviewer comments included in decision letter(s). The following individuals involved in review of your submission have agreed to reveal their identity: Miguel Angel Cevallos (Reviewer #1)

Transaction Report:

DOI: <https://doi.org/10.1128/msystems.00508-23>

June 27, 2023

Prof. Jian-Hua Liu
South China Agricultural University College of Veterinary Medicine
Wushan, Tianhe District
Guangzhou
China

Re: mSystems00508-23 (The origin and evolution of IncF33 plasmids based on large-scale datasets)

Dear Prof. Jian-Hua Liu:

Thank you for submitting your manuscript to mSystems. We have completed our review and I am pleased to inform you that, in principle, we expect to accept it for publication in mSystems. However, acceptance will not be final until you have adequately addressed the reviewer comments.

Preparing Revision Guidelines

Please return the manuscript within 60 days; if you cannot complete the modification within this time period, please contact me. If you do not wish to modify the manuscript and prefer to submit it to another journal, please notify me of your decision immediately so that the manuscript may be formally withdrawn from consideration by mSystems.

Sincerely,

Robert Beiko

Editor, mSystems

Journals Department
Reviewer comments:

Reviewer #1 (Comments for the Author):

Gao and coworkers did a nice job describing the evolution of IncF33 plasmids. They carefully analyzed the structure of a data set of 103 complete plasmids, identified the plasmid backbone, and analyzed variation within the structure of the plasmids. I liked the work done, and I'm sure that the paper will make a nice contribution to the field. I have a request: It is important to indicate which plasmids form cointegrates. Also, indicate if these cointegrates consist of two or more plasmids and mention which plasmid components are in these cointegrates. Finally, it is important to know if the replication modules (different from IncF33) have a narrow or wide host range. A table will be more than enough.

Reviewer #2 (Comments for the Author):

Major comments:

1. The authors analyzed the variable regions of IncF33 plasmids and detailed the genome characters of different clusters. It is worth discussing the potential influence of diverse variable regions on plasmids and bacteria. In addition, among all clusters, cluster A contained the largest number of plasmids but only 8 plasmids. It is necessary to consider adjusting the parameters of CD-Hit to make this analysis more representative.
2. The paragraphs in the part of results and discussion need to be well-organized to present the results logically.
3. Line 351, There are two different types of cointegrates of IncF33 plasmids with other types of plasmids, and the author stated the second way resulted from various plasmid fragments insertion in the IncF33 variable regions was more common. However, is it possible that the first way fusion of IncF33 was underestimated, because you excluded a considerable proportion of plasmids carrying blaKPC genes plasmids in the analysis process? (Line 400 "Since plasmids carrying blaKPC genes were formed by cointegrating IncF33 and R-type plasmids and were not typical IncF33 plasmids, the strains carrying the blaKPC genes were excluded from this study.")
4. As for the clustering analysis of variable region sequence, six clusters containing 26 plasmids were emphasized. However, a majority of 59/65 clusters were not further mentioned. What does the author think about these scattered clusters for which containing only 1-2 samples?
5. How were core genes and varied region defined? The relevant methods are not described.
6. The authors should have a paragraph to describe the limit of this study in the part of results and discussion.

Minor comments:

1. Line 51, 74, 84, 101, 133, 193, "domestic" should be changed into "China".
2. Line 40, 224, 235, 379, 557, 565, "IncF2" is usually written as "IncFII".

In this manuscript, Gao et al. analyzed the origin and evolution of IncF33 plasmids based on a large number of genomes. It is always interesting and essential to study the characteristics and evolution of transferable plasmids carrying antibiotic resistance determinants. Overall, the analysis is thorough and present comprehensive knowledge about IncF33 plasmids. I have a few comments as described below.

Major comments:

1. The authors analyzed the variable regions of IncF33 plasmids and detailly described the genome characters of different clusters. It is worth discussing the potential influence of diverse variable regions on plasmids and bacteria. In addition, among all clusters, cluster A contained the largest number of plasmids but only 8 plasmids. It is necessary to consider adjusting the parameters of CD-Hit to make this analysis more representative.
2. The paragraphs in the part of results and discussion need to be well-organized to present the results logically.
3. Line 351 , There are two different types of cointegrates of IncF33 plasmids with other types of plasmids, and the author stated the second way resulted from various plasmid fragments insertion in the IncF33 variable regions was more common. However, is it possible that the first way fusion of IncF33 was underestimated, because you excluded a considerable proportion of plasmids carrying blaKPC genes plasmids in the analysis process? (Line 400 “Since plasmids carrying blaKPC genes were formed by cointegrating IncF33 and R-type plasmids and were not typical IncF33 plasmids, the strains carrying the blaKPC genes were excluded from this study.”)
4. As for the clustering analysis of variable region sequence, six clusters containing 26 plasmids were emphasized. However, a majority of 59/65 clusters were not further mentioned. What does the author think about these scattered clusters for which containing only 1-2 samples?
5. How were core genes and varied region defined? The relevant methods are not described.
6. The authors should have a paragraph to describe the limit of this study in the part of results and discussion.

Minor comments:

1. Line 51, 74, 84, 101, 133, 193, “domestic” should be changed into “China”.
2. Line 40, 224, 235, 379, 557, 565, “IncF2” is usually written as “IncF II”.

Response to Reviewers

Reviewer comments:

Reviewer #1 (Comments for the Author):

Comments: Gao and coworkers did a nice job describing the evolution of IncF33 plasmids. They carefully analyzed the structure of a data set of 103 complete plasmids, identified the plasmid backbone, and analyzed variation within the structure of the plasmids. I liked the work done, and I'm sure that the paper will make a nice contribution to the field. I have a request: It is important to indicate which plasmids form cointegrates. Also, indicate if these cointegrates consist of two or more plasmids and mention which plasmid components are in these cointegrates. Finally, it is important to know if the replication modules (different from IncF33) have a narrow or wide host range. A table will be more than enough.

Answer: Thank you for your positive feedback and valuable suggestions. We agree that indicating which plasmids form cointegrates and providing details on the plasmid components involved will help readers better understand the evolution of IncF33 plasmids. To address your request, we have prepared Table S4 providing information on IncF33 cointegrate plasmids.

Reviewer #2 (Comments for the Author):

Comment 1: The authors analyzed the variable regions of IncF33 plasmids and detailly described the genome characters of different clusters. It is worth discussing the potential influence of diverse variable regions on plasmids and bacteria. In addition, among all clusters, cluster A contained the largest number of plasmids but only 8 plasmids. It is necessary to consider adjusting the parameters of CD-Hit to make this analysis more representative.

Answer: Thank you for your thoughtful comments and valuable suggestions on our work. We have discussed the potential influence of diverse variable regions on plasmids and bacteria (Lines 344-347). Regarding CD-Hit parameters, we tested

multiple cutoffs previously. By iteratively testing different parameters, we found the optimal value was 0.8 (i.e., 0.8 produced the most representative clustering with well-separated clusters), while lower cutoffs of 0.6-0.7 resulted in imprecise clustering (i.e., distinct variable regions grouped together). We therefore do not recommend lowering the cutoff further as it may introduce inaccuracies.

Comment 2: The paragraphs in the part of results and discussion need to be well-organized to present the results logically.

Answer: We agree with the reviewer. We have reorganized the Results and Discussion sections to make it more logical and clearer.

Comment 3: Line 351, There are two different types of cointegrates of IncF33 plasmids with other types of plasmids, and the author stated the second way resulted from various plasmid fragments insertion in the IncF33 variable regions was more common. However, is it possible that the first way fusion of IncF33 was underestimated, because you excluded a considerable proportion of plasmids carrying *bla_{KPC}* genes plasmids in the analysis process? (Line 400 "Since plasmids carrying *bla_{KPC}* genes were formed by cointegrating IncF33 and R - type plasmids and were not typical IncF33 plasmids, the strains carrying the *bla_{KPC}* genes were excluded from this study.")

Answer: Thank you for pointing out this important issue regarding cointegrate plasmids. We agree that excluding plasmids carrying *bla_{KPC}* genes may make the first way fusion of IncF33 underestimated. However, in this study, we mainly focused on typical IncF33 plasmids. We have reclassified the 292 plasmids (Lines 102-111), categorizing the *bla_{KPC}*-positive cointegrated plasmids into a third class (Type III), and re-described the formation and proportions of Type I and Type II plasmids (Lines 365-370). At the same time, in the newly added Limitations section (Lines 388-394), we also acknowledged the limitation of excluding the *bla_{KPC}*-positive cointegrated

plasmids.

Comment 4: As for the clustering analysis of variable region sequence, six clusters containing 26 plasmids were emphasized. However, a majority of 59/65 clusters were not further mentioned. What does the author think about these scattered clusters for which containing only 1-2 samples?

Answer: Thank you for raising this important issue. In our analysis, clusters with only 1-2 plasmids were indeed scattered and underrepresented. We did analyze plasmids in these clusters, and found most of them represented intermediate evolutionary forms. However, due to the need to summarize variable region characteristics and the word limitation, we could not provide detailed descriptions of these underrepresented clusters, and only focused on major clusters. We added brief descriptions of these scattered clusters in the Results and Discussion sections (Lines 336-340).

Comment 5: How were core genes and varied region defined? The relevant methods are not described.

Answer: We have added the definition of core genes and variable regions in the Materials and Methods section (Lines 446-452).

Comment 6: The authors should have a paragraph to describe the limit of this study in the part of results and discussion.

Answer: We have added a new paragraph in the Results and Discussion sections (Lines 388-394) to describe the limitations of our study.

Minor comments: Line 51, 74, 84, 101, 133, 193, "domestic" should be changed into "China". Line 40, 224, 235, 379, 557, 565, "IncF2" is usually written as "IncFII".

Answer: We have changed "domestic" into "China" throughout the manuscript, and "IncF2" has been modified to the more accurate designation "F2:A-B-".

August 2, 2023

Prof. Jian-Hua Liu
South China Agricultural University College of Veterinary Medicine
Wushan, Tianhe District
Guangzhou
China

Re: mSystems00508-23R1 (The origin and evolution of IncF33 plasmids based on large-scale datasets)

Dear Prof. Jian-Hua Liu:

Your manuscript has been accepted, and I am forwarding it to the ASM Journals Department for publication. For your reference, ASM Journals' address is given below. Before it can be scheduled for publication, your manuscript will be checked by the mSystems production staff to make sure that all elements meet the technical requirements for publication. They will contact you if anything needs to be revised before copyediting and production can begin. Otherwise, you will be notified when your proofs are ready to be viewed.

If you would like to submit a potential Featured Image, please email a file and a short legend to msystems@asmusa.org. Please note that we can only consider images that (i) the authors created or own and (ii) have not been previously published. By submitting, you agree that the image can be used under the same terms as the published article. File requirements: square dimensions (4" x 4"), 300 dpi resolution, RGB colorspace, TIF file format.

We recognize that the video files can become quite large, and so to avoid quality loss ASM suggests sending the video file via <https://www.wetransfer.com/>. When you have a final version of the video and the still ready to share, please send it to mSystems staff at msystems@asmusa.org.

Sincerely,

Robert Beiko
Editor, mSystems

Journals Department
E-mail: mSystems@asmusa.org